# SD3.5-FLASH: DISTRIBUTION-GUIDED DISTILLATION OF GENERATIVE FLOWS

## ABSTRACT

We present SD3.5-Flash, an efficient few-step distillation framework that brings high-quality image generation to accessible consumer devices. Our approach distills computationally prohibitive rectified flow models through a reformulated distribution matching objective tailored specifically for few-step generation. We introduce two key innovations: "timestep sharing" to reduce gradient noise and "split-timestep fine-tuning" to improve prompt alignment. Combined with comprehensive pipeline optimizations like text encoder restructuring and specialized quantization, our system enables both rapid generation and memory-efficient deployment across different hardware configurations. This democratizes access across the full spectrum of devices, from mobile phones to desktop computers. Through extensive evaluation including large-scale user studies, we demonstrate that SD3.5-Flash consistently outperforms existing few-step methods, making advanced generative AI truly accessible for practical deployment.

## 1 INTRODUCTION

Today's best image generation models are trapped in datacenters. While rectified flow models achieve unprecedented quality, their computational demands – 25+ steps, 16GB+ VRAM, 30+ seconds per image – make them inaccessible to everyday devices. We bridge this gap, enabling high-quality generation from mobile phones to gaming desktops.

Timestep distillation offers a path forward. Approaches like distribution matching can reduce step counts in multi-step diffusion inference, but the core challenge emerges from how distribution matching operates in few-step flow distillation. Standard approaches (Yin et al., 2024a; Starodubcev et al., 2025) require re-noising samples on trajectory end-points to compute distribution divergences at various noise levels. This re-noising alters the flow trajectory, resulting in unreliable velocity predictions and corrupted gradient estimates. In few-step regimes, this problem becomes particularly pronounced as errors cannot be corrected through subsequent iterations, causing systematic quality collapse. Additionally, the severe capacity constraints imposed by few-step distillation forces models to sacrifice prompt-image alignment as they struggle to maintain both aesthetic quality and semantic fidelity. Recent image generation pipelines (Starodubcev et al., 2025; Stability AI, 2024) improve prompt-image alignment with parameter-heavy text encoders (Raffel et al., 2020) which further reduces generation efficiency.

We propose SD3.5-Flash, a few-step rectified flow model that enables high-quality image generation (see Fig. 1) on consumer hardware. To train for improved aesthetic quality with few-step flow distillation, we introduce timestep sharing: computing distribution matching with student trajectory samples rather than estimates to random trajectory points. This provides stable gradient signals for known noise levels and reliable flow predictions on the ODE trajectory, improving training stability and consequently model performance.

We also introduce Split-timestep fine-tuning which addresses the prompt alignment challenge by temporarily expanding model capacity during training. Instead of forcing compressed parameters to handle both aesthetic quality and semantic fidelity simultaneously, we branch our model for different timestep ranges before merging them into a unified checkpoint.

To truly deliver on the "flash" promise, we implement pipeline optimizations extending beyond our core algorithmic innovation. We restructure text encoders with optional (T5-XXL) and necessary (CLIP-L/G) components by exploiting encoder dropout pre-training, and apply quantization schemes

from 16-bit to 6-bit precision that balance memory footprint against inference speed. The result is model variants that democratize access across the full spectrum of devices from mobile to desktop, with tailored configurations for each computational tier (see Fig. 2).

Our contributions are aimed to improve accessibility to few-step image generation models through: (i) timestep sharing that provides stable gradients by leveraging intermediate trajectory information, (ii) split-timestep fine-tuning that resolves the capacity-quality tradeoff during distillation, and (iii) comprehensive pipeline optimizations that enable practical deployment on a diverse range of commodity hardware. Through extensive evaluation including large-scale user studies, we demonstrate that our approach consistently outperforms existing methods across diverse hardware configurations while maintaining the quality standards of much larger, slower models.

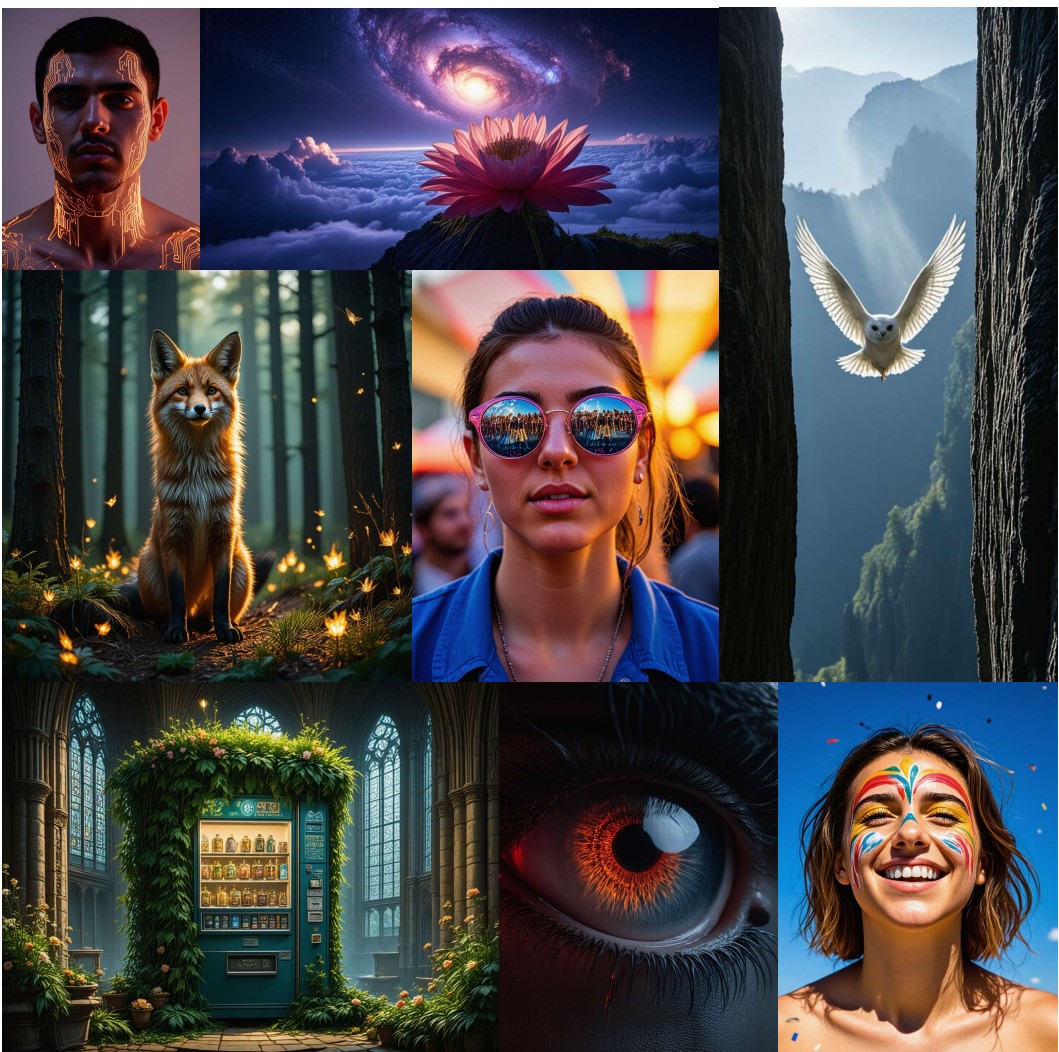

Figure 1: **First Look**: High-fidelity samples (prompts and more samples in appendix) from our 4-step model demonstrate exceptional prompt adherence and compositional understanding. Our method excels where previous distillation approaches often struggle: anatomy and multi-object composition – all while running on affordable consumer hardware.

## 2  RELATED WORKS

Diffusion-based generative models (Ho et al., 2020; Podell et al., 2023) are inherently slow due to their iterative nature, starting from a base distribution (e.g., Gaussian noise) and gradually denoising it to realistic samples. Skip-step schedulers (Song et al., 2020a) accelerate diffusion inference by reducing the number of inference timesteps with deterministic sampling (Karras et al., 2022) while distillation techniques (Luhman & Luhman, 2021; Ren et al., 2024; Chen et al., 2024a; Meng et al., 2023; Kohler et al., 2024) learn a more efficient denoising trajectory.

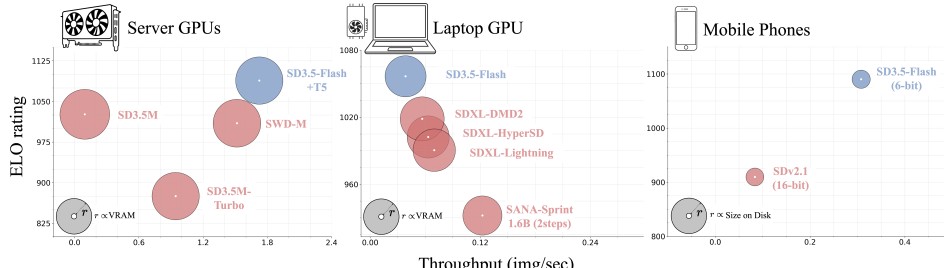

Figure 2: **SD3.5-Flash suite**: We introduce the SD3.5-Flash suite of models, preferred by users over all other models at a variety of consumer compute budgets while offering comparable latency and memory requirements. Bubble size indicates VRAM occupied and pipeline size on disk for gpus and mobile devices respectively. We compute ELO ratings by assessing generated image quality via human rankings for different models.

Trajectory preserving distillation distills a multi-step teacher into a few-step student by aligning the student and teacher trajectories (Salimans & Ho, 2022; Lin et al., 2024) and fine-tuning the student to skip steps progressively. The student learns to mimic an approximation of the teacher's trajectory in fewer steps than the teacher. **Progressive Distillation** of this nature, however, cannot learn extreme low-step (*e.g.* two-step) inference (Lin et al., 2024) due to approximation errors.

Other approaches like discrete (Song et al., 2023; Song & Dhariwal, 2023; Chen et al., 2024b) and continuous time (Lu & Song, 2024; Chen et al., 2025) **Consistency Models** involve learning to jump directly to trajectory endpoints or intermediate points (Kim et al., 2023; Ren et al., 2024) using a more efficient path from noise to data. This improves one-step inference quality while supporting iterative refinement of generated samples through a self-consistency property. Alternately, recent works inspired by Score Distillation Sampling (Poole et al., 2022; Wang et al., 2023), train the student network by **Score Matching** (Song et al., 2020b) of teacher and student distributions (Yin et al., 2024b;a; Starodubcev et al., 2025; Nguyen & Tran, 2024; Dao et al., 2024). Different from these approaches, Insta-Flow (Liu et al., 2023) fine-tunes score based generative models in a rectified flow setting for efficient inference. SWD (Starodubcev et al., 2025) applies DMD for scale wise distillation in a rectified flow setup.

Approaches like progressive distillation, consistency distillation, and score matching are generally unstable or inadequate by themselves and have been supplemented with adversarial techniques in recent works like SDXL-Lightning (Lin et al., 2024), Hyper-SD (Ren et al., 2024) and DMD-2 (Yin et al., 2024a). This adversarial objective is generally optimized by comparing fake samples generated by the few-step student with real (Yin et al., 2024a) or synthetic samples (Sauer et al., 2024a) from the multi-step teacher in a generator discriminator setting. Recent work (Sauer et al., 2024a; Lin et al., 2024) also reformulates this GAN setup to use the teacher as a discriminative feature extractor, for enhancing discriminator quality at no additional cost. This allows for adding multiple lightweight discriminator heads (Chen et al., 2024a) to construct multi-discriminator setups (Sauer et al., 2022; 2023) which offer richer generator updates and training stability through diverse adversarial feedback in GANs. Nitrofusion (Chen et al., 2024a) demonstrates that multi-discriminator adversarial setups are enough without supplementary objectives for stable one-step distillation from low-step models.

Orthogonal to distillation, some methods look to reduce diffusion model parameters (Zhao et al., 2024; Liu et al., 2024; Li et al., 2023; Choi et al., 2023) to further bring down inference cost both in terms of speed and compute. Since attention units take up a large chunk of compute, particularly in recent Diffusion Transformer (DiT) architectures, a majority of works focus on removing (Zhao et al., 2024) or replacing (Liu et al., 2024) them with more efficient alternatives. Separate from the diffusion model itself, the generation pipeline involves the text encoder (Raffel et al., 2020; Radford et al., 2021) for conditional context and the VAE (Kingma et al., 2013) for decoding latent space samples to image space. Some works (Zhao et al., 2024; Bohan, 2024) also focus on optimizing the VAE based latent decoding (denoised latent → image ) by replacing the VAE with a lighter and more efficient decoders.

## 3 BACKGROUND

**Flow matching**. Diffusion Models (Ho et al., 2020; Rombach et al., 2022; Podell et al., 2023) are a family of generative models that learn a (Gaussian) noise to data trajectory and iteratively follow

it to generate media with sampled noise. This trajectory from noise to data is typically modelled as the solution to a Stochastic Differential Equation (SDE) in score-based generative frameworks (Song et al., 2020a), and can be reformulated as an Ordinary Differential Equation (ODE) known as the probability flow ODE (PF-ODE in Song et al. (2020b); Karras et al. (2022)). Diffusion models in score based generative frameworks learn a score function — the gradient of the log probability density — by training a neural network to estimate it at various noise levels along the trajectory. The update direction can be defined as :

$$\mathrm{d}x_t = \left[ \mu(x_t, t) - \frac{1}{2}\sigma(t)^2 \nabla \log p_t(x_t) \right] \mathrm{d}t \tag{1}$$

where $\nabla \log p_t(x_t)$ is referred to as the score function of $p_t(x_t)$ and is parameterised by a neural network as $s_\theta(x_t, t)$ and in a PF-ODE (Karras et al., 2022), $\mu(x_t, t) = 0$. In contrast, flow matching (Lipman et al., 2022; Esser et al., 2024) models define a separate class of generative methods that directly learn an ODE-based mapping without relying on an underlying SDE. These models parameterise a velocity field that transports samples from noise to data along the ODE-defined trajectory. The update direction with flow matching changes to $\mathrm{d}x_t = v_t(x_t)\mathrm{d}t$ where the velocity $v_t(x_t)$ is parameterised by a network as $v_\theta(x_t, t)$. In rectified flow pipelines (Liu et al., 2022) like SD3.5 Medium (Stability AI, 2024), samples are noised following a straight path between the data distribution and standard normal $\mathcal{N}(\mathbf{0}, \mathbf{I})$ as $x_t = (1-t)x_0 + t.\epsilon$

**Distribution Matching Distillation**. DMD (Yin et al., 2024b) proposes the distillation of a multi-step teacher $G$ into a distilled single-step student $G_\theta$ by matching the student distribution $p_{\text{fake}}$ with that of the teacher $p_{\text{real}}$. Given a sample $x = G_\theta(z)$ where $z \sim \mathcal{N}(0, \mathbf{I})$ this distribution match is calculated as the Kullback-Leibler (KL) divergence:

$$D_{\text{KL}}(p_{\text{fake}}||p_{\text{real}}) = -\mathbb{E}_{x \sim p_{\text{fake}}} \Big( \log p_{\text{real}}(x) - \log p_{\text{fake}}(x) \Big) \tag{2}$$

However, using this divergence directly as loss is not possible as the probability densities are generally intractable. Since only the gradient of this loss is needed, this can be circumvented, by substituting in score function $s(x) = \nabla_x \log p(x)$ and computing the loss gradient as

$$\nabla_\theta \mathcal{L}_{\text{DMD}} = -\mathbb{E}_{x \sim p_{\text{fake}}} \left( (s_{\text{real}}(x) - s_{\text{fake}}(x)) \frac{\mathrm{d}G_\theta}{\mathrm{d}\theta} \right) \tag{3}$$

To obtain these scores, generated samples $x_0$ are re-noised up-to timestep $t$ as $x_t = \sqrt{\alpha_t}x + \sqrt{1-\alpha_t}\epsilon$. Then the score is computed from the denoising signal of the pre-trained diffusion models as $s_{\text{real}}(x_t, t)$ for teacher score and $s_{\text{fake}}(x_t, t)$ for student score where $s_{\text{fake}}(x_t, t) = -\frac{x_t - \alpha_t G_\theta(x_t, t)}{\sigma_t^2}$ from the student $G_\theta$. Since the few-step models work only on a subset of timesteps, a multi-step proxy model is maintained that monitors the distribution of the few-step model and acts as a surrogate student score estimator. To stabilise this pipeline, $\mathcal{L}_{\text{DMD}}$ is accompanied by regression loss, calculated as the MSE between images generated by the student and the teacher starting from the same noise. DMD2 (Yin et al., 2024a) proposes updating the student proxy $G_\phi$ with a biased schedule to improve stability without introducing this regression loss and supplements $\mathcal{L}_{\text{DMD}}$ with an adversarial objective.

## 4 METHODOLOGY

### 4.1 TRAJECTORY GUIDANCE

For stable pre-training of our 4-step student network, we use a trajectory guidance objective $\mathcal{L}_{\text{TG}}$. For timesteps $t \in [0, 1]$ on the teacher model's trajectory, we subsample points $t_i^s$ which coincide with the student trajectory (*i.e.* $i \in [1, 4]$ for 4-step model) and calculate the trajectory guidance objective as:

$$\mathcal{L}_{\text{TG}} = \sum_i \|t_i^s(G_\theta(x_{t_i^s}, t_i^s) - \int_{t_i^s}^{t_{i-1}^s} v_{\text{real}}(x_t, t)\mathrm{d}t)\|^2 \tag{4}$$

where $v_{\text{real}}$ corresponds to the velocity predictor teacher model and $G_\theta$ is the student being trained.

### 4.2 DISTRIBUTION MATCHING IN FLOW MODELS

We refine our pre-trained student using the DMD objective in Eq. (3) that computes the gradient for the KL-divergence between teacher and student distributions with the proxy ($v_{\text{fake}}$). We align the distributions of the proxy and the student, to enable accurate representation of student distribution in

$\mathcal{L}_{\text{DMD}}$ by finetuning on generated student samples $x_0$. Particularly end-point estimates $x_0$ are noised to $x_t$ and flow-matching loss is computed as $\mathcal{L}_{\text{FM}} = ||v_{\text{target}} - v_{\text{fake}}(x_t, t)||_2^2$, where $v_{\text{target}}$ is from added noise. To train student $G_\theta$ for timestep $t_i$ ($i \in [2, 4]$), we disable gradients and use the student itself to generate upto $t_{i-1}$. Unlike Yin et al. (2024a) we find that starting training directly on slightly noisier samples $x_{t_{i-1}}$ for timestep $t = t_i$ improves performance compared to training on sample $x_{t_i}$. After training stabilises, we switch back to training on $x_{t_i}$ for timestep $t = t_i$, similar to "backward simulation" proposed by Yin et al. (2024a).

**Timestep Sharing.** The DMD objective in Eq. (3), requires noising samples to $x_t$ from $x_0$ to compute the real and fake scores $s_{\text{real}}(x_t, t)$ and $s_{\text{fake}}(x_t, t)$ respectively. In score based models, this is done by adding random noise to samples which is already part of the denoising loop. However, pre-trained flow based models have matching image noise pairs and adding random noise for reaching timestep $t$ can create noisy gradient updates. We simplify the training objective and prevent noise addition by sharing DMD timesteps with those from the few-step denoising schedule.

Specifically, we evaluate the KL divergence gradient not by re-noising from trajectory endpoints (*i.e.* $x_0$ to $x_t$ in Eq. (3)), but by simply using partially denoised samples ($x_{t^s}$) on the student trajectory for velocity estimation. Intuitively, we calculate the score for assumed "pseudo" $x_0$ that is noised to $x_{t_i^s}$ instead of estimating $x_0$ itself (see Fig. 3). This reduces low quality gradients from poor $x_0$ estimation from noisy timesteps (at $t \approx 1$). Consequently, this forces us to share distribution matching timesteps with the student trajectory timesteps $t_i^s$, instead of random $t$ in Eq. (3). While this does result in less variation in timesteps (using only few timesteps from student

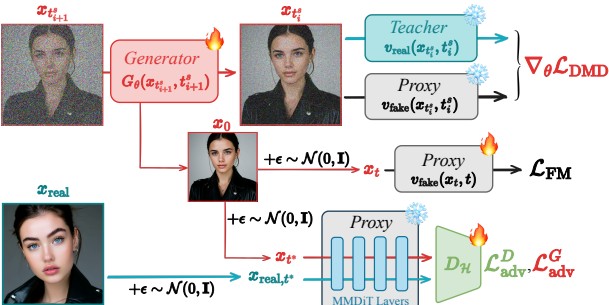

Figure 3: **Training Pipeline**: We train $G_\theta$ with the distribution matching objective $\nabla_\theta \mathcal{L}_{\text{DMD}}$ and adversarial objective $\mathcal{L}_{\text{adv}}^G$. The proxy student $v_{\text{fake}}$ that is used to compute $\nabla_\theta \mathcal{L}_{\text{DMD}}$ is trained with the standard flow matching objective $\mathcal{L}_{\text{FM}}$ and the discriminator for the adversarial objective is trained with $\mathcal{L}_{\text{adv}}^D$.

trajectory), we find it improves image composition and generation quality (see Sec. 5.5).

**Split-Timestep Fine-Tuning.** Timestep distillation often weakens the correspondence between text prompts and generated outputs (Sauer et al., 2024a). To counteract this, we design split-timestep fine-tuning, inspired by previous works that employ diffusion models for multi-task learning (Ham et al., 2025; Ma et al., 2025). We first duplicate the pretrained model into branches, $M_1$ and $M_2$ and train them on disjoint timestep ranges $t_1 \in (0, 500]$ and $t_2 \in (500, 1000]$ respectively, to increase effective model capacity. During fine-tuning, each branch uses an exponential moving average with a decay of $\beta = 0.99$ to stabilise and keep weights close to the original checkpoint. After convergence, we fuse the branches by weight interpolation, selecting a $3 : 7$ ratio ($M_1 : M_2$) to maximise text-prompt alignment as measured with GenEval (Ghosh et al., 2023). We perform split timestep fine-tuning only for training our four step model where we observe a distinct jump in model performance.

### 4.3 ADVERSARIAL LOSS

Similar to prior works (Chen et al., 2024a; Yin et al., 2024a), we use an adversarial objective where the proxy student $v_{\text{fake}}$ acts as a feature extractor to obtain discriminator features. This allows us to perform adversarial training on the flow latent space as opposed to the image space in (Sauer et al., 2024b). For extracting features using $v_{\text{fake}}$, we noise samples $x_0$ to pre-defined noise levels at timesteps $t^* \in [0, 1]$ and extract intermediate outputs from $v_{\text{fake}}(x_{t^*}, t^*)$ at multiple layers as feature maps. Timesteps $t^*$ are well distributed in $[0, 1]$ to capture both coarse-grained features ($t^* \approx 1$) and fine-grained features ($t^* \approx 0$). We train MLP discriminator heads $D_\mathcal{H}$ on top of these features for real/fake prediction where synthetic samples generated by the teacher model are used as "real" data. Similar to NitroFusion (Chen et al., 2024a), we periodically refresh our discriminator heads by re-initializing their weights to reduce overfitting. We use the standard non saturating GAN objective to train the discriminator heads and the generator $G_\theta$:

$$\mathcal{L}_{\text{adv}}^D = \mathbb{E}_{x_{t^*} \sim p_{\text{real}, t^*}} \log D(x_{t^*}) - \mathbb{E}_{x_{t^*} \sim p_{\text{fake}, t^*}} \log D(x_{t^*}), \quad \mathcal{L}_{\text{adv}}^G = -\mathbb{E}_{x_{t^*} \sim p_{\text{fake}, t^*}} \log D(x_{t^*}) \quad (5)$$

where the discriminator heads $D_\mathcal{H}$ (Fig. 3) and the feature extractor are collectively referred to as $D$.

## 4.4 TWO STEP AND FOUR STEP GENERATION

For training a two step generator, we progressively distill a multi-step teacher down to a four step student and continue training it towards two step inference. We start by initializing our teacher, student and the proxy student with pre-trained weights from the multi-step teacher. Next, we perform two stages of training, where we (i) pre-train the student model with $\mathcal{L}_{\text{TG}}$ where the model is optimized to replicate the teacher trajectory in few-steps. (ii) In the second stage, we minimize the KL divergence of teacher and student distributions $\mathcal{L}_{\text{DMD}}$ supplemented with an adversarial objective from our multi-head discriminator. The first stage of training helps to align teacher and student trajectories and speeds up training of the next stage considerably. The second stage constructs sharp features and detailed images. We use the trained four step model as our pre-trained checkpoint to distill down to two step following the second stage of our training pipeline. In here, we also use a MSE objective between gram matrices (Gatys et al., 2016) of features from samples of teacher and student models.

## 4.5 PIPELINE OPTIMIZATION

We perform inference optimization on top of the Stable Diffusion 3.5 pipeline. This pipeline consists of three text encoders (CLIP-L (Radford et al., 2021), CLIP-G (Radford et al., 2021), and T5-XXL (Raffel et al., 2020)) besides the MM-DiT diffusion model (Stability AI, 2024), and a VAE (Kingma et al., 2013). Of these, T5-XXL is the largest component, accounting for the bulk of peak VRAM usage and inference time. The full distilled model in 16-bit precision requires 18 GiB of GPU memory—beyond the reach of most consumer cards. To bring this down, we quantize the MM-DiT diffusion model to 8-bit and leverage encoder dropout pre-training in SD3.5 to substitute T5-XXL with null embeddings. This brings our memory requirement down to just about 8 GiB. To truly support edge devices like phones and tablets, we use CoreML on Apple Silicon to quantize our 8-bit model down to 6-bit (Fig. 2). Specifically for this quantization, we rewrite operations like RMSNorm to better preserve precision on the Apple Neural Engine. We summarise the results of our optimzation in Tab. 1, and highlight less than 10s latency on devices like iPhone (video in supplementary zip) and iPad. We include more details on memory performance tradeoff in Fig. 8.

Table 1: **Inference latency**: Comparing inference latency of SD3.5-Flash models for different devices with VRAM / unified memory below device names.

| Model | Steps | Resolution | Latency (in seconds) | | | |
|---|---|---|---|---|---|---|
| | | | RTX 4090 24 GB | M3 MBP 32 GB | M4 iPad 8 GB | A17 iPhone 8 GB |
| SD3.5-Flash 16-bit (w T5-XXL) | 4 | 1024 px | 0.58 | 18.65 | – | – |
| | | 768 px | 0.34 | 8.21 | – | – |
| | | 512 px | 0.19 | 3.74 | – | – |
| SD3.5-Flash 8-bit (w/o T5-XXL) | 4 | 1024 px | 0.61 | 14.08 | – | – |
| | | 768 px | 0.35 | 6.32 | – | – |
| | | 512 px | 0.22 | 2.97 | – | – |
| SD3.5-Flash 6-bit (w/o T5-XXL) | 4 | 1024 px | – | 13.43 | – | – |
| | | 768 px | – | 6.26 | 6.44 | 8.32 |
| | | 512 px | – | 3.12 | 2.62 | 3.25 |

## 5 EXPERIMENTS

### 5.1 IMPLEMENTATION DETAILS

**Dataset and Training.** Following previous works (Chen et al., 2024a; Sauer et al., 2024a), we use synthetic samples for training our model as they offer high prompt coherence and are consistent in quality. For our training data, we generate synthetic samples using the SD3.5 Large (8B) model over 32 timesteps and a CFG scale of 4.0. We pre-train for 2K iterations and then train the 4-step and 2-step model for 1200 iterations each, using the 2.5B SD3.5M as teacher. The 2-step model starts training from a 4-step intermediate checkpoint. We present more training details in the appendix.

**Baselines.** For comparisons, we look at **DMD2** (Yin et al., 2024a), **Hyper-SD** (Ren et al., 2024), **SDXL-Turbo** (Sauer et al., 2024b), **Nitrofusion** (Chen et al., 2024a) and **SDXL-Lightning** that are trained from **SDXL** (Podell et al., 2023) as the teacher network. DMD2 distils SDXL by matching the distributions of the teacher and the student with the gradient of a KL divergence objective. Hyper-SD performs consistency distillation with trajectory guidance and uses human feedback learning (Xu et al., 2023) for improving performance. SDXL-Turbo demonstrates adversarial distillation in the rich semantic space of Dino-V2 (Oquab et al., 2023), decoding latents to images throughout

training. SDXL-Lightning also uses adversarial distillation, but relaxes mode coverage for the student with a mix of conditional and unconditional objectives in the discriminator. Nitrofusion stabilises adversarial distillation with a multi-discriminator setup and a periodic discriminator refresh, training on SDXL-DMD2 and SDXL-HyperSD. Improving upon SDXL and **SDv2.1** (Rombach et al., 2022), recent models like **SD3.5** (Stability AI, 2024) and **SANA** (Xie et al., 2025) offer better generation quality and higher prompt adherence by adopting rectified flow pipelines for faster convergence. **SWD** (Starodubcev et al., 2025) distils SD3.5M by training a scale wise network, optimized with a distribution matching objective. **SANA-Sprint** (Chen et al., 2025) uses continuous-time consistency distillation (Song et al., 2023) to distil SANA to 1, 2, and 4-step models. We also include comparisons with **SD3.5M-Turbo** released by TensorArt Studios (TensorArt Studios, 2025) as an stand-alone checkpoint on top of SD3.5M. We do not compare with large models like SD3.5 Large (8B) and Flux.1-dev (Black Forest Labs, 2024) (12B) which are difficult to fit into consumer grade hardware.

## 5.2 QUALITATIVE COMPARISONS

With T5     Without T5

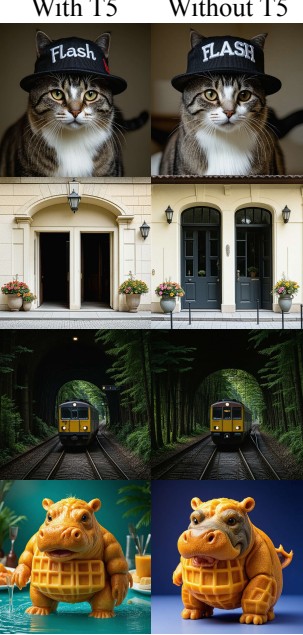

We include qualitative comparisons of our model (**SD3.5-Flash 16-bit + T5**) with other few-step generation pipelines like SANA-Sprint1.6B, NitroFusion, SDXL-DMD2 and SDXL-Lightning in Fig. 5, and additional comparisons (including SWD) in the appendix. 4-step results from SDXL-DMD2 (Yin et al., 2024a), SDXL-Lightning f(Lin et al., 2024) and NitroFusion (Chen et al., 2024a) show poor prompt alignment and composition in complex prompts involving human interaction. SDXL-Lightning (Lin et al., 2024) generates smooth images lacking sharpness and low in detail, and sometimes generates artifacts (*e.g.* two corgis on sofa in last row, last column). SDXL-DMD2 (Yin et al., 2024a) and NitroFusion (Chen et al., 2024a) (distilled from SDXL-DMD2) generate better texture but similarly perform worse in composition and result in artifacts (second row, cat on the book and first row, three owls). Comparatively, our method (4-step) consistently generates high quality images and outperforms other 4-step pipelines in generation fidelity considerably. In 2-step pipelines, we compare with SANA-Sprint 1.6B (Chen et al., 2025). SANA-Sprint (Chen et al., 2025) generates more details but with inconsistent style, sometimes generating stylistic images (first and third column) without style prompt. SANA-Sprint (Xie et al., 2025) also generates smudged facial features in non close-up environments (see fourth row). Our 2-step method outperforms SANA-sprint in generation fidelity, but lags behind (missing book in third row and artifacts in fourth row) our 4-step model. We also provide examples of our 4-step 16-bit model with and without T5 in Fig. 4.

Figure 4: **Removing T5**: 4 step quality with and w/o T5 (prompts in appendix)

## 5.3 USER STUDY

We conduct a user study based on image quality and prompt alignment with 124 annotators to evaluate images generated with 4 different seeds. For generating samples, we use a diverse curated set of 507 prompts consisting of expert-designed prompts and a subset of Parti prompts (Yu et al., 2022). For each generated sample, 3 users vote on two images from two different methods, rating them on visual quality and image-prompt correlation (prompt adherence). From our user studies (in Fig. 6), we find SD3.5-Flash outperforms other few-step models and even our 50 step teacher in image quality. For prompt-adherence, the difference is marginal ($< \pm 1.6\%$) across all methods (more in appendix). We also compare select competitors against each other to compute ELO scores (see Fig. 2). In all compute scenarios our models appear on the top of the ELO ladder demonstrating high quality image generation across a variety of compute budgets.

## 5.4 QUANTITATIVE COMPARISONS

We conduct extensive quantitative validation (in Tab. 2) by generating 30K samples for captions from the COCO dataset (Lin et al., 2014), where we use metrics like **ImageReward** (Xu et al., 2023) **CLIPScore** (Radford et al., 2021), **FID** (Heusel et al., 2017), and **Aesthetic Score** (Schuhmann et al., 2022) to quantify generation performance. ImageReward (IR) and Aesthetic Score (AeS) are human preference metrics and are trained to reflect human preferences on image quality. Metrics like CLIPScore and FID are computed for quantifying text alignment and similarity to real images

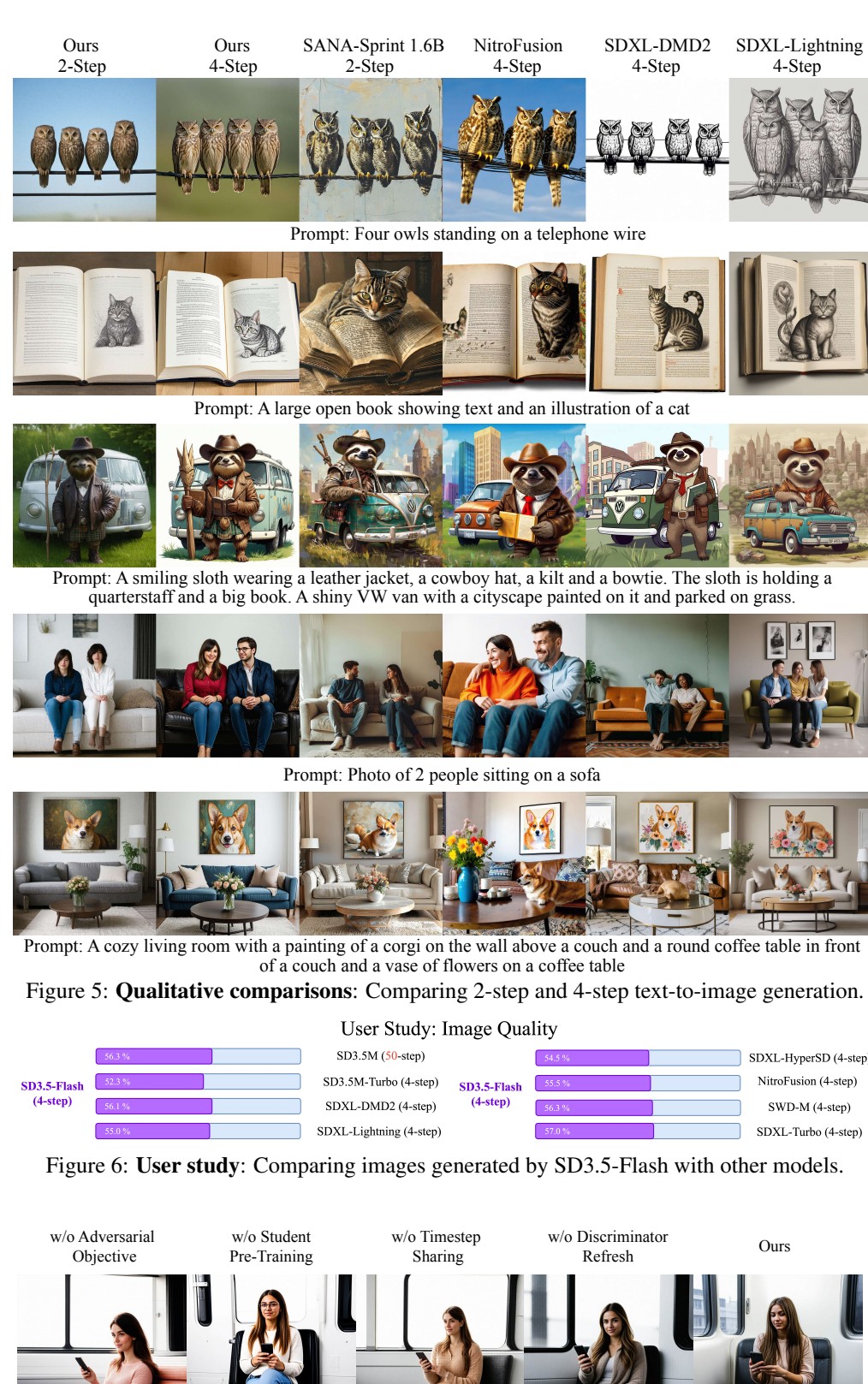

Figure 5: **Qualitative comparisons**: Comparing 2-step and 4-step text-to-image generation.

Figure 6: **User study**: Comparing images generated by SD3.5-Flash with other models.

Figure 7: **Ablative study**: Demonstrating the importance of each component in our training pipeline.

Table 2: **Quantitative comparison**: Comparison with other models on automated metrics. Models that use SD3.5M are coloured in green.

| Methods | Steps | Latency (s) (↓) | Peak VRAM (GiB) (↓) | CLIP (↑) | FID (↓) | AeS (↑) | IR (↑) | GenEval (↑) |
|---|---|---|---|---|---|---|---|---|
| SDXL (Podell et al., 2023) | 50 | 5.81 | 8.95 | 31.65 | 14.72 | 6.32 | 0.72 | 0.54 |
| SD3.5M (Stability AI, 2024) | 50 | 10.58 | 19.47 | 32.00 | 20.06 | 5.99 | 0.91 | 0.64 |
| SDXL-Turbo (Sauer et al., 2024b) | 4 | 0.43 | 8.95 | 31.67 | 20.76 | 6.19 | 0.84 | 0.56 |
| SDXL-Lightning (Lin et al., 2024) | 4 | 0.43 | 8.96 | 31.25 | 21.48 | 6.48 | 0.74 | 0.54 |
| SDXL-DMD2 (Yin et al., 2024a) | 4 | 0.43 | 8.96 | 31.64 | 16.64 | 6.28 | 0.88 | 0.56 |
| SDXL-HyperSD (Ren et al., 2024) | 4 | 0.45 | 9.32 | 31.59 | 24.01 | 6.67 | 1.05 | 0.56 |
| NitroFusion (Real.) (Chen et al., 2024a) | 4 | 0.43 | 8.96 | 31.28 | 22.66 | 6.41 | 0.91 | 0.55 |
| SWD-M (Chen et al., 2024a) | 4 | 0.66 | 17.88 | 32.00 | 25.90 | 6.37 | 1.12 | 0.72 |
| SD3.5M-Turbo (w CFG) (TensorArt Studios, 2025) | 4 | 1.06 | 17.59 | 31.16 | 26.14 | 5.86 | 0.30 | 0.54 |
| SD3.5-Flash 16-bit (w T5-XXL) | 4 | 0.58 | 17.58 | 31.65 | 29.80 | 6.38 | 1.10 | 0.70 |
| SD3.5-Flash 16-bit (w/o T5-XXL) | 4 | 0.55 | 8.71 | 31.63 | 28.65 | 6.39 | 1.08 | 0.68 |
| SD3.5-Flash 8-bit (w 8-bit T5-XXL) | 4 | 0.66 | 11.17 | 31.64 | 29.99 | 6.37 | 1.10 | 0.70 |
| SD3.5-Flash 8-bit (w/o T5-XXL) | 4 | 0.61 | 6.61 | 31.62 | 28.84 | 6.39 | 1.08 | 0.68 |
| SDXL-Turbo (Sauer et al., 2024b) | 2 | 0.30 | 8.95 | 31.73 | 22.65 | 6.22 | 0.81 | 0.55 |
| SDXL-Lightning (Lin et al., 2024) | 2 | 0.30 | 8.96 | 31.18 | 21.99 | 6.40 | 0.66 | 0.69 |
| SDXL-DMD2 (Yin et al., 2024a) | 2 | 0.31 | 8.96 | 31.63 | 16.67 | 6.28 | 0.87 | 0.56 |
| SDXL-HyperSD (Ren et al., 2024) | 2 | 0.32 | 9.32 | 31.97 | 27.26 | 6.50 | 1.12 | 0.55 |
| NitroFusion (Real.) (Chen et al., 2024a) | 2 | 0.30 | 8.96 | 31.47 | 20.83 | 6.36 | 0.91 | 0.55 |
| SANA-Sprint 0.6B (Chen et al., 2025) | 2 | 0.22 | 8.2 | 31.39 | 24.99 | 6.54 | 0.98 | 0.77 |
| SANA-Sprint 1.6B (Chen et al., 2025) | 2 | 0.24 | 10.17 | 31.43 | 23.10 | 6.61 | 1.01 | 0.73 |
| SD3.5-Flash 16-bit (w T5-XXL) | 2 | 0.39 | 17.58 | 31.82 | 29.37 | 6.32 | 1.00 | 0.70 |
| SD3.5-Flash 16-bit (w/o T5-XXL) | 2 | 0.36 | 8.71 | 31.73 | 28.88 | 6.36 | 0.94 | 0.67 |
| SD3.5-Flash 8-bit (w 8-bit T5-XXL) | 2 | 0.44 | 11.17 | 31.81 | 29.43 | 6.31 | 1.00 | 0.70 |
| SD3.5-Flash 8-bit (w/o T5-XXL) | 2 | 0.40 | 6.61 | 31.73 | 28.92 | 6.35 | 0.94 | 0.67 |

respectively. CLIPScore is measured as the similarity between text prompts and generated images in CLIP ViT-B/32 (Kolesnikov et al., 2021) semantic space. FID (Heusel et al., 2017) is calculated as the distance between distributions of generated and real images (from COCO here) in the Inception-V3 (Szegedy et al., 2016) feature space. We also include comparisons on the **GenEval** (Ghosh et al., 2023) score where images of specific objects are generated in different settings and evaluated with an object detection framework for identifying text-to-image alignment. We compare against all baselines and competitors with these metrics along with their corresponding **Latency** as the time taken to generate a sample on a RTX 4090 GPU with 16-bit float precision (BF16) unless otherwise specified. From Tab. 2, we find that our method offers competitive performance for text to image generation compared to recent works like SDXL-DMD2 and NitroFusion, while surpassing the teacher model SD3.5M in metrics like GenEval, AeS and IR. Despite being calculated on the same COCO-30K dataset, we note that our FID is worse off while other metrics have competitive scores. We attribute this to FID difference between teachers SDXL and SD3.5M themselves, noting that SD3.5M-Turbo and SWD trained from SD3.5M have worse FID on average.

## 5.5 ABLATIVE STUDIES

We conduct ablative experiments (Fig. 7) by distilling SD3.5M (16-bit 4-step) without individual components in our pipeline, showing their importance for generation fidelity. Particularly, we distill the model: (i) **w/o Adversarial Objective**: where we do not use GAN training for guiding generation, (ii) **w/o Pre-Training**: Where we do not pre-train the student generator $G_\theta$, (iii) **w/o Timestep Sharing**: Where we use random timestep $t$ for $x_t$ in $\mathcal{L}_{DMD}$ instead of those on the student trajectory, and (iv) **w/o Discriminator Refresh**: Where the discriminator heads are not periodically re-initialised to correct overfitting. We train the ablation students for the same iterations as our student model. We find that removing the adversarial objective destabilises training. resulting in poor generation quality. Without pre-training, colour and composition are impacted the most. Training without timestep sharing also results in poor texture, colour, and composition. Finally, without discriminator refresh we find minor compositional errors and over smooth images.

## 6 CONCLUSION

As in all distillation processes, we trade-off some aspect of quality and diversity with inference speed in complex generation tasks. We find that removing T5 for faster inference with lower memory also makes it difficult to construct complex compositions from worse conditional context (Fig. 4). However, these limitations are not unique to our method and are a natural consequence of approximating diffusion trajectories with low-step models. Despite them, we find our 4-step model offers up-to $\sim 18\times$ speed-up on the teacher and surpasses it in average performance on large scale user studies with various levels of prompt complexity. **We include a summary video in the supplementary zip for a quick overview.**

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

# A   APPENDIX

## A.1   TRAINING

We distill **SD3.5 Medium** (SD3.5M) from 50 steps down to 4 steps and 2 steps. For our multi-head discriminator setup, we extract features from layers 3,4,5,6,8,10 and 11 of proxy SD3.5M student with MM-DiT architecture. Each of these heads consists of 8 MLP layers where in the first 4 layers, patch features are individually attended to, and then combined to compute discriminator logits in the next 4 layers. We use LayerNorm and SiLU activation units in between MLP layers. At each iteration, discriminator heads have a probability $p = 0.005$ of getting re-initialised to reduce overfitting and are updated with the proxy student network ($v_{\text{fake}}$) 10 times for every single generator ($G_\theta$) update. In the pre-training stage we train $G_\theta$ for $2K$ iterations with a learning rate of $1e - 6$, optimizer AdamW, and an effective batch size of 140 per GPU over 8 H100s taking 17 hours. For stage two, we use an effective batch size of 80 (per GPU) and train $v_{\text{fake}}$, $G_\theta$ and discriminator network ($D$) with learning rates $1e - 6$, $5e - 6$, and $5e - 5$ respectively (with AdamW) for 800 iterations, taking 6 hours on 8 H100s. We train on top of the 4-step model for 2-step generation with stage 2 of our training pipeline, training for 1200 iterations (9 hours on 8 H100s) . For both our 4-step and 2-step model, we distribute denoising timesteps uniformly over $[0, 1]$. For split-timestep fine-tuning, we further train our 4-step checkpoint for 400 iterations (4 hours on 8 H100s).

## A.2   QUANTIZATION TRADEOFF

We provide a visual analysis of the memory v/s performance tradeoff for quantizing SD3.5-Flash on a M3 Macbook Pro with 32 GiB of memory (Fig. 8).

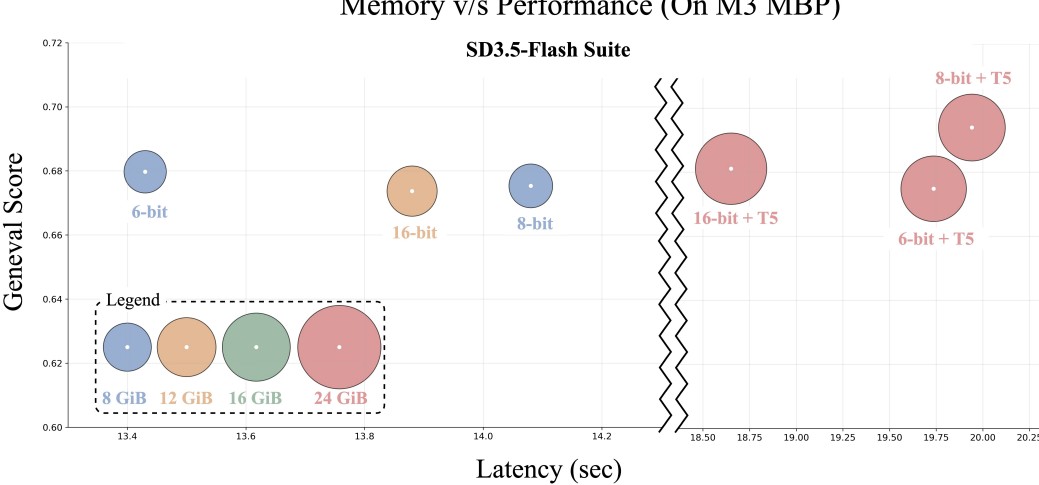

Figure 8: **Latency v/s GenEval**: Comparison of Latency and GenEval scores for 4-step inference pipelines

## A.3   USER STUDY ANALYSIS

We include results from our user study for prompt adherence in Fig. 9 and perform an analysis of the 507 prompts used (Fig. Fig. 6 and Sec. 5.3) in Fig. 10. Specifically, we use GPT-4 to categorise prompts into pre-determined labels and to score prompt complexity particularly for image generation. Through our ablations, we found it beneficial to disentangle image quality and prompt alignment preferences, because otherwise users tend to conflate the two factors and we obtain a less clear signal. Specifically, when participants were asked to choose the better image in terms of aesthetics, the prompt was hidden. Conversely, for the prompt alignment task, participants were instructed to focus solely on alignment with the prompt and disregard image quality. While this setup increases the cost of the study, we adopted it to ensure clearer results. We also include a screenshot of the user interface in Fig. 11 and Fig. 12 for the image quality and prompt alignment tasks. User studies are performed

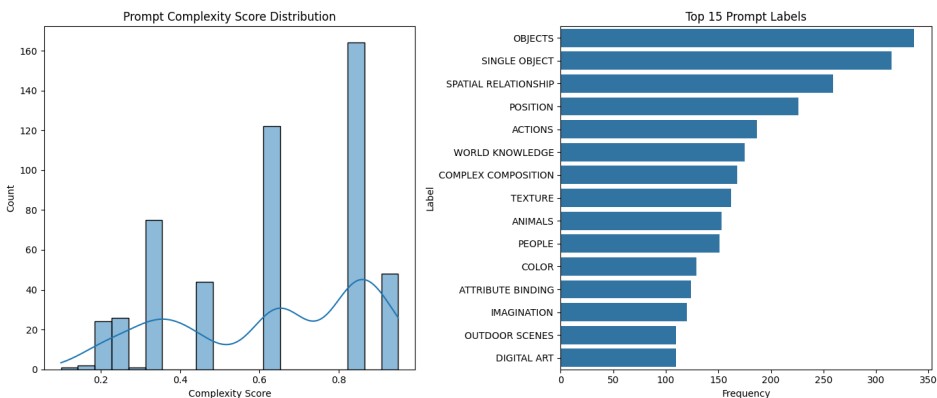

Figure 9: **Prompt Adherence:** User ratings for prompt adherence demonstrated by different models.

with candidates who have prior experience in ranking generated images, and as such do not require any explicit instructions, after multiple rounds of quality check.

Figure 10: **User study prompt analysis:** Left: Our prompt set covers a wide distribution of complexity as a function of prompt length and categories. Right: Top 15 prompt labels and their frequency.

## A.4 Additional qualitative analysis

We include more images from our 4-step model in Fig. 15 and comparisons of our 4-step and 2-step results with those from other models in Figs. 13 and 14.

## A.5 Prompt list

We include all prompts used to generate Figs. 1, 4 and 15 here:

**Fig. 1** From top to bottom, left to right:

- Portrait of a man with glowing circuitry embedded in his skin, neutral expression
- A radiant galaxy seen from a cliff above the clouds, with a giant flower blooming from the mountaintop in the foreground
- A white owl soaring vertically between two cliff walls with sunlight streaming from above
- A majestic red fox standing upright on its hind legs in a glowing forest, fireflies swirling around
- Portrait of a person with holographic sunglasses reflecting a carnival scene in vivid daylight
- A vending machine overgrown with flowers and ivy, humming softly in the center of a ruined cathedral with stained glass light pouring in
- Extreme close-up of a cybernetic eye with rotating mechanical parts and glowing red highlights
- Portrait of a smiling person with multicolored face paint under a clear blue sky, confetti falling around

**Fig. 4** From top to bottom:

**1. Which image has better visual quality?**

Also use both and None, if applicable.

○ Left Image
○ Right Image
○ Both
○ None

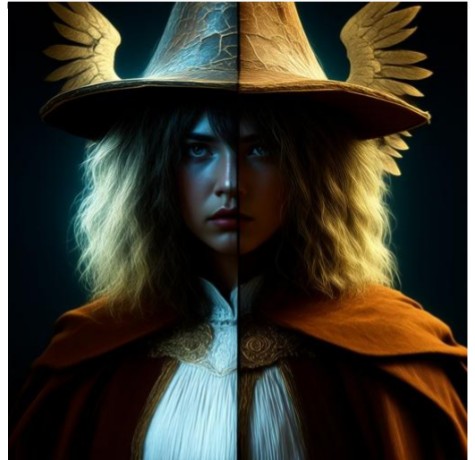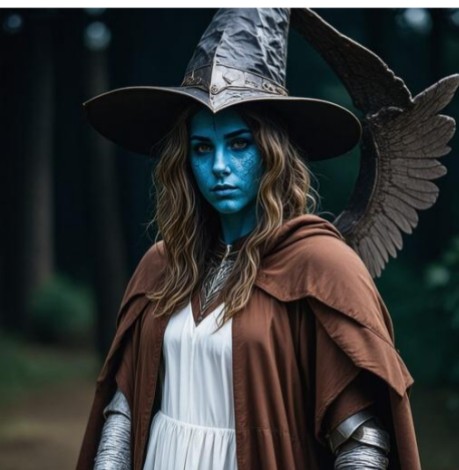

Figure 11: **User Study for image quality:** Users are asked to select their preferred image only based on image quality

**1. Which image better matches the given prompt?**
*Please ignore visual aesthetics. Focus only on how well the image aligns with the prompt's content.*

○ Left Image
○ Right Image
○ Both images match equally well
○ Neither image matches the prompt

**Prompt:** upper body, split screen, splitscreen, outdoors, night, . , IncrsXLRanni, wavy hair, cracked skin, blue skin, colored skin, extra faces, white dress, witch hat, brown cloak, , . MaleniaDef, armor, cape, winged helmet, brown dress, covered eyes, helmet

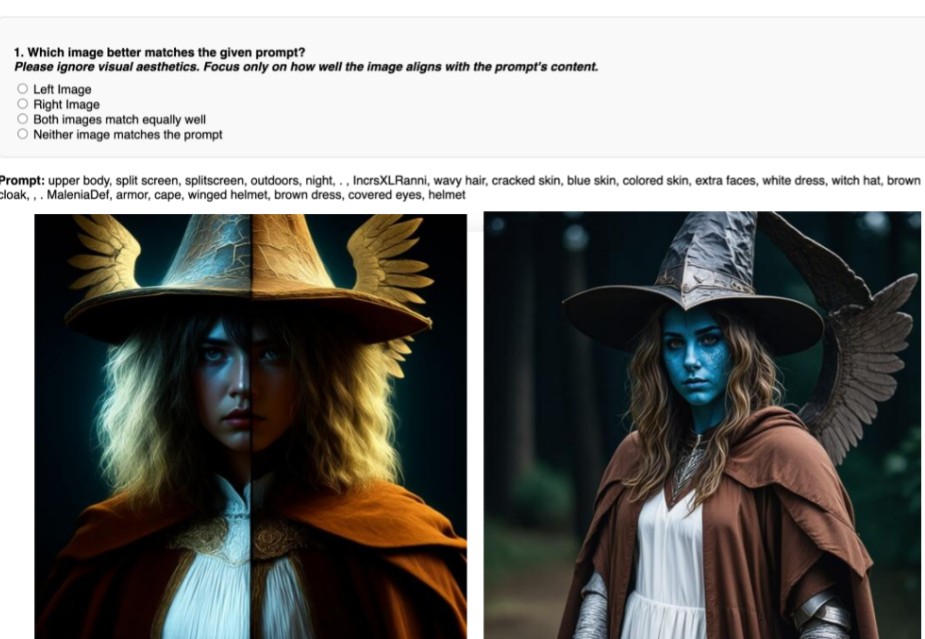

Figure 12: **User Study for prompt alignment:** Users are asked to select their preferred image only based on prompt alignment, while discarding image aesthetic

| Ours
4-Step | NitroFusion
4-Step | SDXL-DMD2
4-Step | SDXL-Lightning
4-Step | SWD
4-Step |
|---|---|---|---|---|

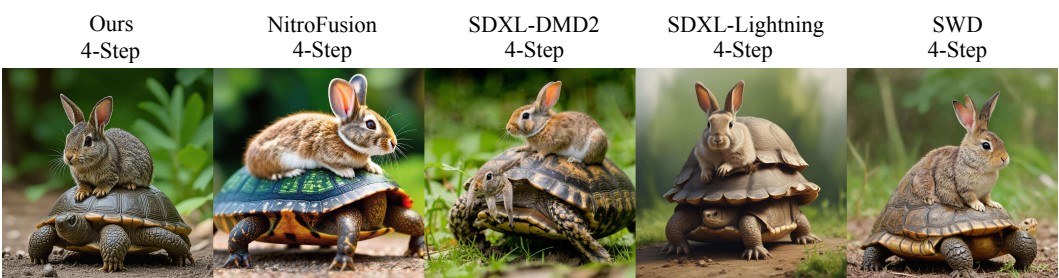

Prompt: a rabbit sitting on a turtle's back.

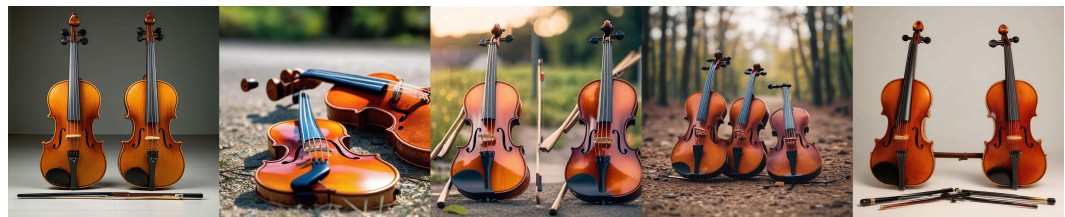

Prompt: two violins standing up with their bows on the ground in front of them.

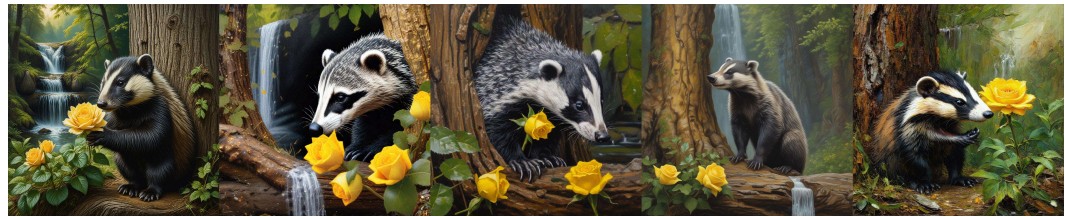

Prompt: A richly textured oil painting of a young badger delicately sniffing a yellow rose next to a tree trunk. A small waterfall can be seen in the background.

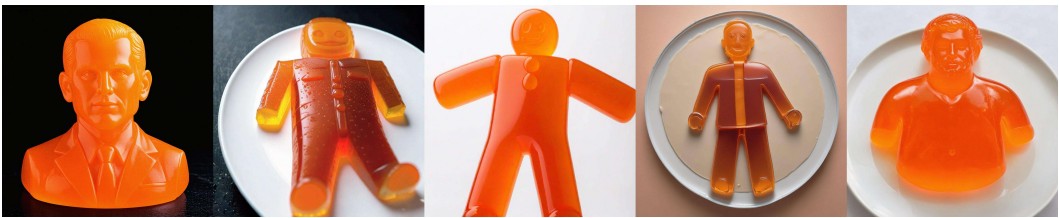

Prompt: orange jello in the shape of a man

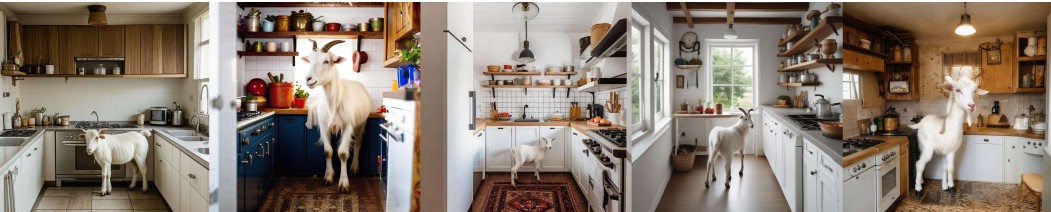

Prompt: a small kitchen with a white goat in it.

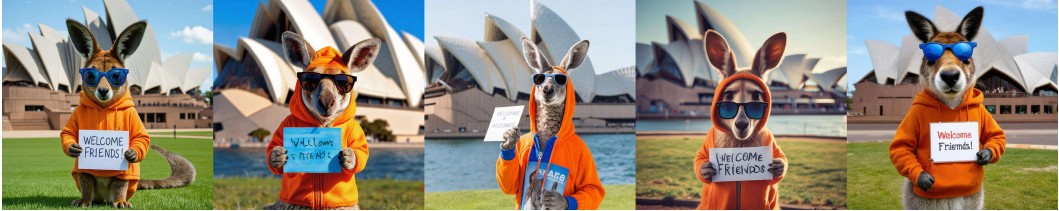

Prompt:A portrait photo of a kangaroo wearing an orange hoodie and blue sunglasses standing on the grassin front of the Sydney Opera House holding a sign on the chest that says Welcome Friends.

Figure 13: **Qualitative Comparison:** Additional qualitative comparisons with other four step distilled models.

|      Ours      |      Ours      | SANA-Sprint 1.6B | NitroFusion | SDXL-DMD2 | SDXL-Lightning |
|    2-Step     |    4-Step     |    2-Step    |   4-Step   |   4-Step   |   4-Step   |

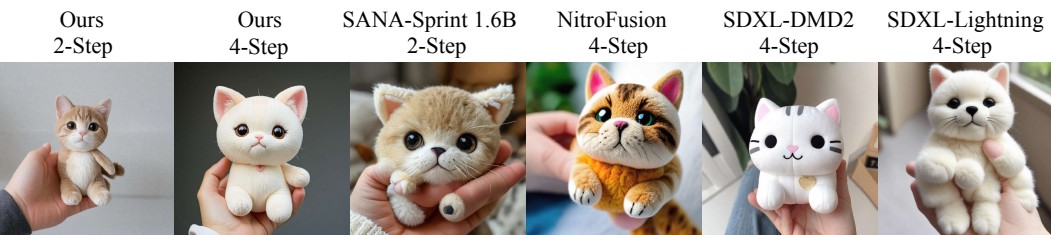

Prompt: Close up of a hand holding a cute cat plushie

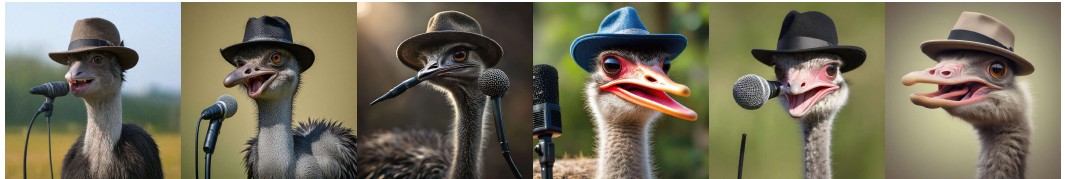

Prompt: A photograph of an ostrich wearing a fedora and singing soulfully into a microphone

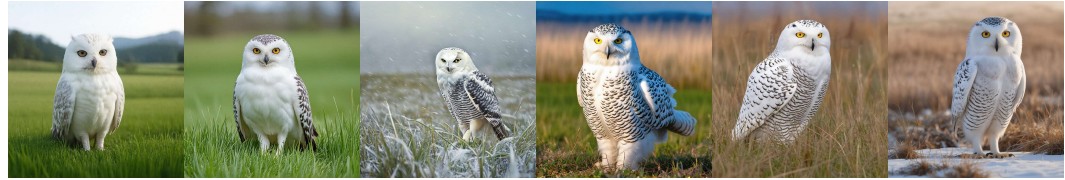

Prompt: A snowy owl standing in a grassy field

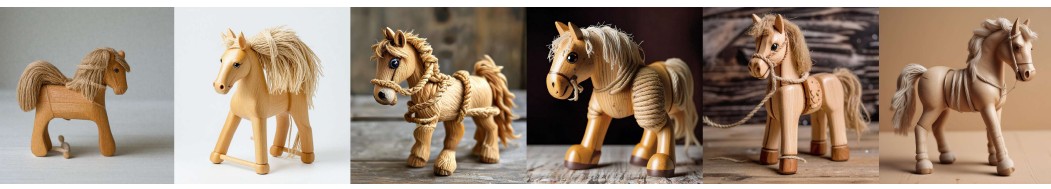

Prompt: A wooden toy horse with a mane made of rope

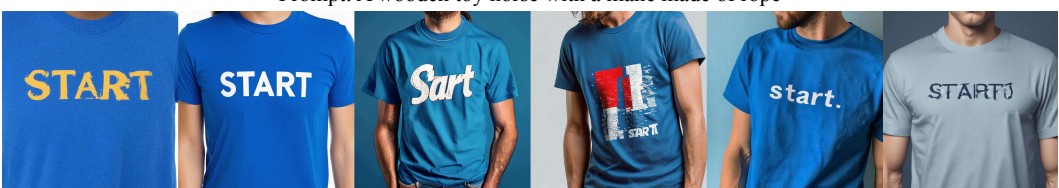

Prompt: The word 'START' on a blue t-shirt

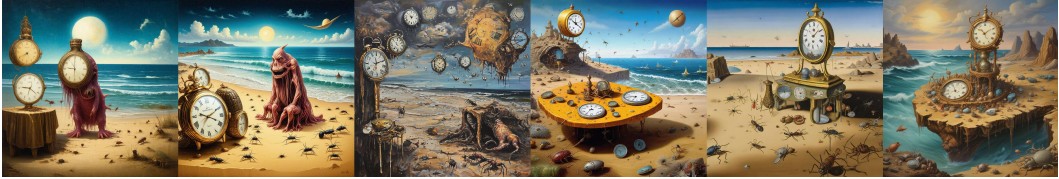

Prompt: An oil surrealist painting of a dreamworld on a seashore where clocks and watches appear to be inexplicably limp and melting in the desolate landscape. a table on the left, with a golden watch swarmed by ants. a strange fleshy creature in the center of the painting

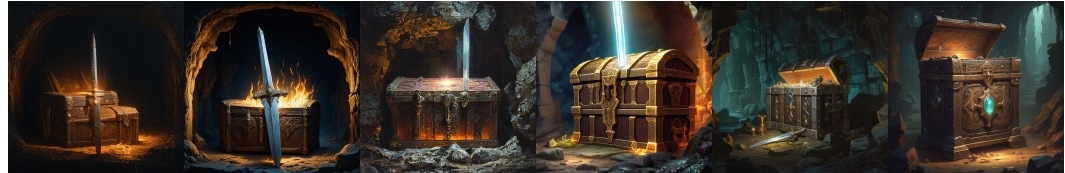

Prompt: A painting of an ornate treasure chest with a broad sword propped up against it, glowing in a dark cave

Figure 14: **Qualitative Comparison:** Additional qualitative comparisons with other few-step distilled models.

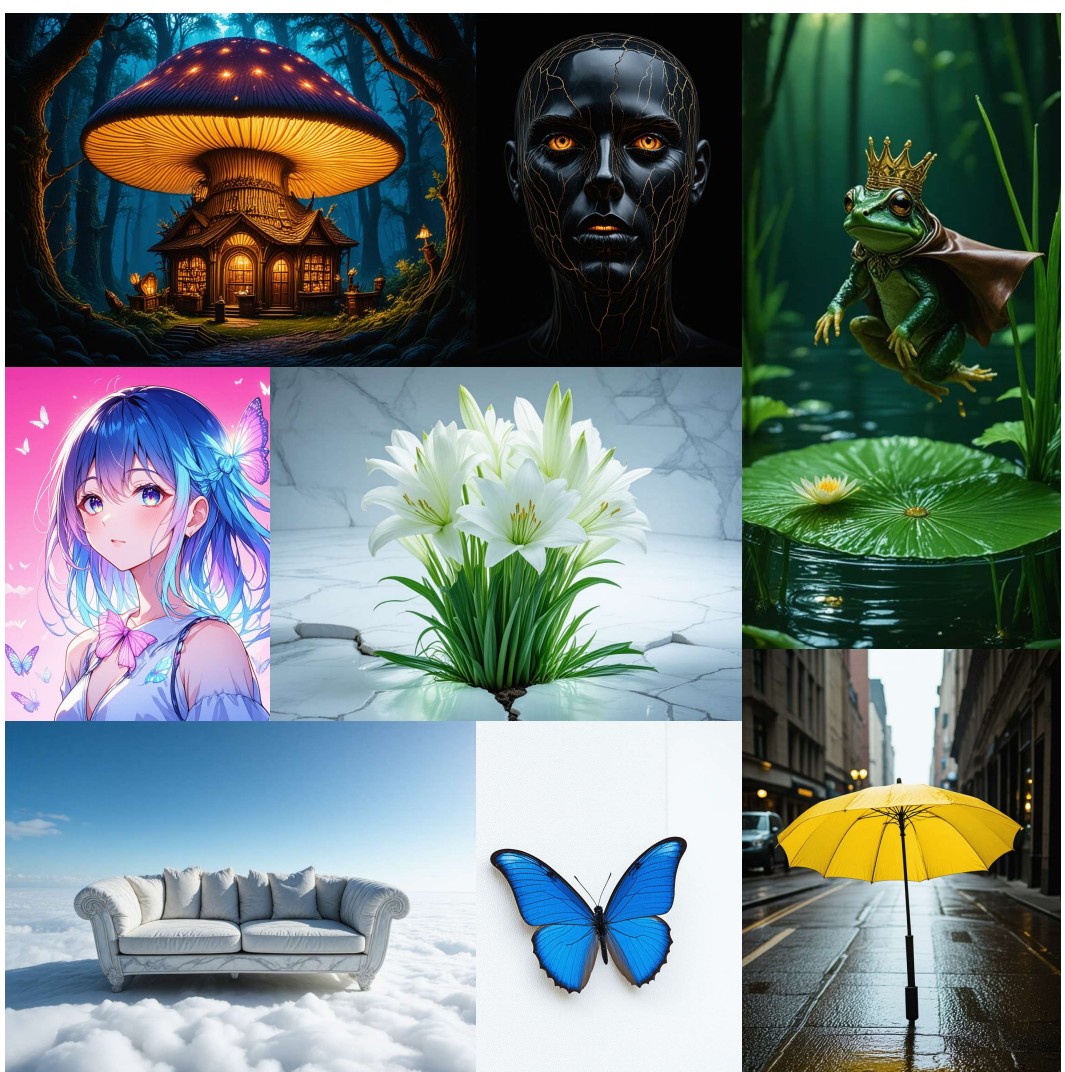

Figure 15: **Qualitative Comparison:** Additional high fidelity results from our 4-step model in different aspect ratios.

- A photo of a cat with a hat that says "Flash" in white letters. Artistic style.
- A building wall and pair of doors that are open, along with vases of flowers on the outside of the building.
- A passenger train traveling through a tunnel covered with a forest.
- A whimsical and creative image depicting a hybrid creature that is a mix of a waffle and a hippopotamus. This imaginative creature features the distinctive, bulky body of a hippo, but with a texture and appearance resembling a golden-brown, crispy waffle. The creature might have elements like waffle squares across its skin and a syrup-like sheen. It's set in a surreal environment that playfully combines a natural water habitat of a hippo with elements of a breakfast table setting, possibly including oversized utensils or plates in the background. The image should evoke a sense of playful absurdity and culinary fantasy.

**Fig. 15** From top to bottom, left to right:

- A fantasy bookstore carved into the glowing cap of a massive mushroom, nestled in a bioluminescent forest at night
- A humanoid face made of smooth obsidian with glowing cracks, set against a black background
- A small frog wearing a crown and cape, leaping up toward a floating lily pad in a glowing swamp
- Close-up of an anime girl with glowing rainbow hair flowing in the wind, surrounded by neon butterflies under a pink sky
- A bouquet of paper-white lilies growing from a crack in an endless marble floor, petals emitting a gentle phosphorescent glow that blends into the radiant surroundings
- A sculpted marble sofa hovering above a cloud deck lit by an overexposed noon sun, cushions shimmering like polished alabaster
- A blue butterfly on a white wall
- A vivid yellow umbrella alone in a rainy city street

