# OpenReview forum: "SD3.5-Flash: Distribution-Guided Distillation of Generative Flows"
_ICLR.cc/2026/Conference — ICLR 2026 Conference Withdrawn Submission_

### Official Review · Reviewer_jdQH · 2025-10-27

**Soundness:** 2
**Presentation:** 3
**Contribution:** 2
**Rating:** 2
**Confidence:** 4

**Summary:**

This paper concentrates on two points:

1. a modification of the renoising part of the DMD algorithm
2. a comprehensive lightning pipeline including text encoder dropping, quantization, and diffusion step distillation

**Strengths:**

1. Good presentation
2. Diffusion acceleration is an important topic, and I appreciate the efforts in combining different acceleration methods.

**Weaknesses:**

### Experiments are inadequate to support the proposed methods:
  1. The ablative experiment is qualitative, but numerical results are necessary here to validate the strengths of the modifications
  2. Cross-base-model comparisons are not informative enough. Considering that the proposed method is evolved from DMD/DMD2, a fair comparison with these works (using the same base model and training pipeline) is necessary to substantiate the modifications as improvements.

### Timestep sharing, as the primary original methodological contribution, is not sufficiently explained and validated

If my understanding is correct, in traditional DMD, the noisy sample is obtained by first denoising (moving) the few-step input to the clean level (t=0) using the velocity predicted by the few-step generator, followed by renoising to the target timestep $t'$ with random noise. In the proposed method, the renoised sample with noise level $t'$ is obtained by directly moving the few-step input along the few-step predicted velocity by a certain interval.

I have the following questions:

1. The sentence "Consequently, this forces us to share distribution matching timesteps with the student trajectory timesteps $t^s_i$, instead of random t in Eq. (3)" does not trivially hold to me. Why does this constraint exist? And what would happen if  $t'$ is randomly sampled without constraints?

2. The sentence "adding random noise for reaching timestep t can create noisy gradient updates." is not intuitive to me either. While score-based methods, diffusion models, and flow matching models have their respective backgrounds and mathematical deductions, in practice, they are doing almost the same thing: adding noise to clean data, and making the model predict a target quantity, such as noise, the original clean data, or velocity. Importantly, these quantities are all interchangeable. Therefore, why do flow matching models uniquely suffer from the unstable gradient problem? Do diffusion models suffer from similar issues?

3. In-depth analysis is needed to explain what the unstable gradient problem is and how it negatively affects training. Quantitative experiments are also needed to support its superiority over the existing renoising method.

### Apart from the above limitations, the overall novelty of this work is limited. It is mainly a combination of well-established techniques.

**Questions:**

See weakness

---

> ### Author Response · Authors · 2025-11-13
> **Clarification on Methodology**
>
> We thank the reviewer for their questions. They allow us to clarify the core contribution, as the reviewer’s points seem to stem from a misunderstanding. The very issues the reviewer questions: why ‘_timestep sharing_’ is necessary and why ‘_unstable gradients_’ are a problem, are the technical challenges that motivate our paper.
>
> We point out a fundamental difference between score and flow models: The **deterministic noise-data pairings** in flow models can lead to noise in score estimation during true DMD computation, this deterministic pairing is missing in score models. This constraint is why we must follow the student’s exact trajectory. Our method avoids the **conflicting, unstable objective** that arises from naively applying standard DMD. This core distinction, which is central to our work, appears to have been missed. To be precise on the implementation:
>
> During "backward simulation’' in DMD training, we construct each training sample through denoising from random noise at $t=1000$ using the student being learned $G_\theta$. To train on a sample (for example) at $t=500$, we first denoise (without gradients) from $x_{1000} \sim \mathcal{N}(0,I)$ through intermediate steps $t=1000 \to 750 \to 500$ to obtain $x_{500}$. Then we would perform the noise/velocity estimation as $v = G_\theta(x_{500},t=500)$ while tracking gradients to compute the DMD objective. The true DMD objective uses this noise/velocity prediction to compute $x_0$ directly, re-noising to reach a random timestep $t$ for score estimation ($s_\text{real},s_\text{fake}$). Instead of re-noising to random $t$, we denoise to the next natural step in the student trajectory (_i.e._ $t=250$ here) using estimated velocity $v$ and compute the KL divergence gradient at this step. Hence, we are not just using the timesteps of the student to compute the DMD objective, but following the student's trajectory **picking noisy samples directly from this trajectory** to estimate the DMD objective.
>
> **Note on Ablations:** Ablation images show significant degradation in aesthetic quality, which is why we do not compute FID scores on ablative experiments. Quantitative metrics like FID have been further shown to have poor correlation with human preference which is why the values from these scores can often not reflect true quality. From qualitative ablative studies, we note specific details like (i) w/o adversarial training: blurry image and background, low details and low resolutions, horizontal pattern lines (ii) w/o student pretraining: clear oversaturation in samples, (iii) w/o timestep sharing: lack of details in background and distortion in subject, (iv) w/o discriminator refresh: faint grid lines throughout image (note the similarity to removing adversarial training), and lack of detail in face or background.
>
> **Note on cross-base model comparisons:** We perform comparisons with SWD and SD3.5M Turbo both of which are distilled from SD3.5M. In here, SWD-M is trained with the DMD2 objective (see L136, L330). It is unrealistic for us to compare against all baselines by training them on SD3.5M.

---

### Official Review · Reviewer_xtfv · 2025-10-28

**Soundness:** 3
**Presentation:** 3
**Contribution:** 2
**Rating:** 6
**Confidence:** 3

**Summary:**

This paper introduces SD3.5-Flash, a distillation framework designed to efficiently convert a large, multi-step teacher model (SD3.5 Medium) into a high-quality, few-step (2-4 steps) student model suitable for deployment on consumer hardware. The core methodological contributions are two-fold:
Timestep Sharing: A simple trick that uses the student's intermediate trajectory points for gradient computation, avoiding the instability introduced by re-noising from trajectory endpoints.
Split-Timestep Fine-Tuning: A capacity-enhancement technique where the model is branched and fine-tuned on disjoint timestep ranges before being merged back, aiming to improve prompt alignment.
Through extensive qualitative, quantitative, and user-study-based evaluations, the authors demonstrate that SD3.5-Flash outperforms existing few-step distillation methods in image quality and prompt adherence while achieving significant speed-ups.

**Strengths:**

1. "Split-Timestep Fine-Tuning" is a creative approach to mitigating the capacity bottleneck in distilled models.
2. The work goes beyond a pure algorithmic contribution by integrating model distillation with critical deployment engineering. This holistic view from algorithm to practical deployment is highly valuable, which proves a potentially significant practical impact.
3. Large-scale human evaluation, broad quantitative metrics (FID, CLIPScore, Aesthetic Score, ImageReward), and latency and memory footprint analysis are valuable for other efforts.

**Weaknesses:**

1. The Timestep Sharing operation seems like a simple trick to integrate DMD and other denoising steps.
2. The efficacy and rationality of Split-Timestep Fine-Tuning should be highlighted and analyzed in detail.
3. Although the paper claims the efficiency of SD3.5-Flash on terminal devices like laptops or phones, the method does not design any optimization strategies to enhance this ability. To give convincing proofs, the author should analyze why the proposed model is superior after quantization compared to others.
4. Typos.
5. I suggest the authors release the code and finetuned models, which will be of high significance for the community.

**Questions:**

see Weaknesses

---

### Official Review · Reviewer_2T9T · 2025-10-31

**Soundness:** 2
**Presentation:** 2
**Contribution:** 2
**Rating:** 4
**Confidence:** 3

**Summary:**

SD3.5-Flash is an efficient few-step distillation framework designed to bring high-quality image generation from computationally intensive rectified flow models down to accessible consumer devices. Its foundation is a reformulated distribution matching objective tailored for the few-step paradigm, minimizing the gap between the original model and the distilled generator. The method introduces key innovations: "timestep sharing" to effectively reduce gradient noise and "split-timestep fine-tuning" to significantly improve prompt alignment. Coupled with comprehensive pipeline optimizations like text encoder restructuring, SD3.5-Flash enables both rapid, high-quality synthesis and highly memory-efficient deployment on resource-constrained hardware.

**Strengths:**

1. The paper exhibits outstanding clarity and excellent presentation; the framework is meticulously detailed, and the overall narrative ensures that the complex concepts are highly accessible to the reader.

2. The work is strongly motivated. Few-step distillation represents a crucial research area for rapid generative models, and the goal of enabling high-quality image synthesis on accessible consumer devices holds significant promise for the field's practical application.

**Weaknesses:**

1. Based on the results presented in Table 2, the proposed method appears to show only marginal, if any, advantage in terms of combined efficiency and capability when directly compared to the SANA-Sprint baseline, potentially weakening the claimed technical contribution of the work.

2. Although titled SD3.5-Flash, the core contribution seems presented as a general few-step distillation framework. The paper would be significantly strengthened by demonstrating its applicability and performance across other advanced flow-matching models, such as FLUX, to confirm the true generality of the approach.

**Questions:**

None

---

> ### Author Response · Authors · 2025-11-13
> **Rebuttal**
>
> We thank the reviewer for their appreciation of our paper clarity and the general direction of our work.
>
> 1. When directly compared to SANA-Sprint baselines, we have worse latency from having a larger model and a lower compression factor in our auto-encoder. However, from Table 2, we would like to clarify that quantitative metrics often do not capture core details like aesthetic quality. We conduct large scale human evaluations with $507$ prompts and $4$ seeds on each prompt, where each sample is reviewed by $3$ users. With these human evaluations we construct an Elo table (Fig. 2), the numbers from which are listed here:
>
> | **Server Compute (> 12GB VRAM)** |            |     | **Laptop Compute (8-12GB VRAM)**     |            |     | **Mobile Phones (< 8GB VRAM)**  |            |
> |----------------------------------|------------|-----|--------------------------------------|------------|-----|--------------------------------|------------|
> | Method                           | Elo        |     | Method                               | Elo        |     | Method                         | Elo        |
> | **SD3.5-Flash 16-bit(+T5)**      | **1088.7** |     | **SD3.5-Flash 16-bit (No T5)**       | **1056.6** |     | **SD3.5-Flash 6-bit**          | **1090.2** |
> | SD3.5M                           | 1026.3     |     | SDXL-DMD2                            | 1018.5     |     | SDv2.1                         | 909.8      |
> | SWD-M                            | 1009.8     |     | SDXL-Lightning                       | 1002.3     |     |                                |            |
> | SD3.5M-Turbo                     | 875.2      |     | SDXL-HyperSD                         | 990.5      |     |                                |            |
> |                                  |            |     | **SANA-Sprint 1.6B**                     | 932.1      |     |                                |            |
>
> Separate from this table, we find that our method beats SANA-Sprint 1.6B in pairwise comparisons from user studies with a win rate of $56.1$ over SANA-Sprint's $43.9$ .
>
> 2. The core idea of our paper is to demonstrate a model that can run on hand-held devices like iPhones and iPads (see supplementary zip where we deploy our model on A17 chip iPhone). Unfortunately large image generation models like Flux (with 12B parameters) are difficult to fit into mobile devices with $8$ GB VRAM.

---

### Note · Authors · 2025-11-14

I have read and agree with the venue's withdrawal policy on behalf of myself and my co-authors.